# Changes in Nutritional State and Cardiovascular Parameters in Alimentary Obese Children after a Month-Long Stay in Children’s Treatment Center

**DOI:** 10.3390/children9111610

**Published:** 2022-10-22

**Authors:** Ksenia Budinskaya, Ondřej Pírek, Natálie Rafčíková, Olga Nádeníčková, Kateřina Bednaříková, Hana Hrstková, Petr Dobšák, Zuzana Nováková

**Affiliations:** 1Department of Physiology, Faculty of Medicine, Masaryk University, Kamenice 5, CZ-625 00 Brno, Czech Republic; 2Children’s Hospital Kretin, Kretin 12, CZ-679 62 Kretin, Czech Republic; 3Department of Pediatrics, Institutions Shared with the Faculty Hospital Brno, Jihlavská 20, CZ-625 00 Brno, Czech Republic; 4Department of Sports Medicine and Rehabilitation, St Anne’s Faculty Hospital and Faculty of Medicine, Masaryk University, Pekařská 664/53, CZ-656 91 Brno, Czech Republic

**Keywords:** childhood obesity, anthropometry, pulse wave velocity

## Abstract

Childhood and adolescent obesity has become an important public health issue, as it leads to higher risk of cardio–metabolic, orthopedic, and psychological comorbidities. The aim of this study was to evaluate the changes in nutritional state and cardiovascular system parameters in obese children. Sixty respondents aged 9–17 years with alimentary obesity participated in this research. Anthropometric parameters (body weight (BWT), body mass index (BMI), percentage of body fat (%), waist and hip circumference (WC and HC), waist–hip ratio (WHR)) and cardiovascular parameters (systolic and diastolic blood pressure (SP and DP), cardio-ankle vascular index (CAVI), ankle-brachial index (ABI), pulse wave velocity and its variability (PWV and PWV_V_), and parameters of pulse wave analysis) were measured. Every respondent went through two sets of measurements, the first (I.) after their admission to the children’s hospital and the second (II.) at the end of their one-month-long therapeutic stay. Statistically significant differences between measurements I. and II. were observed in the following parameters: BWT (*p* < 0.01), BMI (*p* < 0.01), WC (*p* < 0.01), HC (*p* < 0.01), DP (*p* < 0.01), PWV (*p* < 0.05), and ABI (*p* < 0.01). The results of this study show that obesity has a mostly negative impact on the cardiovascular health of affected children, with likely negative results in their adulthood.

## 1. Introduction

In recent years, childhood and adolescent obesity has become an important public health issue, not only in the Czech Republic, but also in many other European countries. According to the World Health Organization, more than four million people die each year as a result of being overweight or obese. Moreover, the prevalence of overweight and obesity in both adults and children continually rises; in children and adolescents, it quadrupled between 1975 and 2016 and as of now, it stands at 18% [1]. Childhood obesity is defined as body mass index (BMI) above the 95th percentile for the respective age and sex, and is defined as “alimentary” when it is caused by caloric intake greater than caloric expenditures [2,3]. It leads to increased incidence of cardiovascular risk factors, such as metabolic syndrome, glucose intolerance, hypertension, atherogenic dyslipidemia, and early atherosclerosis.

The development of habits leading to obesity is affected by multiple factors, some of which are uncontrollable, but most of which can be influenced. The most crucial risk factors include changes in the food industry and excessive energy intake, which is primarily caused by energy–dense foods, sugar–loaded drinks, and lack of physical activity. These risk factors, besides other complications, lead to shorter sleep periods, which are also associated with a higher cardio–metabolic risk profile [4].

The higher cardiovascular risk includes, among other diseases, the risk of atherosclerosis. One of the underlying causes of atherosclerosis is arterial stiffening, which is an age–related, complex process of structural changes in the vessel wall. Continual balance between the two main components of the vascular wall—collagen and elastin—ensures its stability. This stability is disrupted by inflammatory processes, which lead to overproduction of atypical collagens and diminution of elastin fibers [5]. Rich perivascular adipose tissue supports inflammation and thus promotes instability inside the vascular wall [6]. These structural changes in the arterial wall lead to increased arterial stiffness and overall rigidity of the arterial wall. These consequences are manifested by increased pulsatility and afterload, and can lead to hypertension, myocardial infarction, heart failure, and other cardiovascular issues [7].

Persistently increased total fat mass in obese children and adolescents is related to greater arterial stiffness [8,9]. We can determine the arterial stiffness by carotid–femoral pulse wave velocity (PWV), which is obtained using applanation tonometry. Applanation tonometry is the gold–standard method in PWV measurement, but using it in clinical practice, especially in children, poses many challenges. However, the PWV may be considered a useful cardiovascular risk indicator.

Knowing the parameters that determine the risks associated with childhood obesity could help find these patients in time, and start their treatment. Early initiation of treatment in adolescents may prevent the development of serious pathologies and convert into long-term vascular benefit.

The aim of this study was to evaluate the changes in nutritional state and parameters of the cardiovascular system which reflect the condition of the central and peripheral parts of the vascular system, as well as diastolic heart function in obese children.

## 2. Materials and Methods

The study was approved by the ethics committee of the Faculty of Medicine, Masaryk University Brno, the Czech Republic, and was conducted according to the principles stated in the Declaration of Helsinki. All respondents and their parents were properly informed about their role in this study, its aim and purposes, and confirmed this by signing an informed consent, which has been archived.

### 2.1. Study Design

This project was a longitudinal cohort study involving children during their stay in the Children’s Hospital, Kretin, in the Czech Republic. All of the measurements were made between May 2018 and October 2019. As requirements for participation in this study, BMI (above the 95th percentile for respective age and sex) and alimentary etiology of obesity were used. Children with other chronic diseases and comorbidities (bronchial asthma, skin diseases, etc.), as well as with a different etiology of obesity, were excluded from this study.

### 2.2. Subjects

Sixty respondents (38 males and 22 females) aged 9–17 years with alimentary obesity participated in this research. The respondents were divided into 3 age-based groups, in order to discover any age-dependent changes in the cardiovascular system: the 9–11 (13 males, 8 females), 12–14 (16 males, 9 females) and 15–17 (9 males, 5 females) years-of-age groups. The sex of the respondents was also considered, as the results of both sexes within every age group were also compared. The basic characteristics of the respondents are presented in Table 1. (The data shown in this table were obtained during the admission examination of the patients at the hospital.)

Every respondent went through two sets of measurements, first (I.) after their admission to the children’s hospital and second (II.) at the end of their one–month long therapeutic stay. For the male vs. female comparison within one age group, lower indices were used to mark the compared measurements. Every respondent had their anthropometric and cardiovascular parameters measured. Body mass index (BMI) and body weight (BWT) values were also compared with publicly available percentile graphs based on the Czech pediatric population [10].

Respondents were measured in the Children’s Hospital, Kretin, during their one–month stay in this medical facility. They underwent weight-loss therapy based on a complex approach, which relied on the cooperation of a physician, nutrition specialist, psychologist, physiotherapist, and fitness coach. The key part of this weight-loss therapy is cognitive-behavioral therapy (CBT), which utilizes the fact that bad nutritional and movement habits are learned, and therefore can be changed using various psychological techniques. During CBT, the children focused their attention on self–monitoring (planning of their calory intake, physical activity, regular weight checking) and real goal setting; they were also trained to identify internal eating cues (craving or emotional stimuli) and to replace them with alternative types of behavior. This therapy starts as a set of short-term habits which should gradually transform into long-term positive behavioral changes [11,12].

### 2.3. Anthropometric Parameters

BWT in kilograms (kg) and the percentage of body fat (%) were measured using the TANITA personal weight (TANITA InnerScanV, BC-601), which relies on the bioimpedance method. Body height (m) of the respondents was measured using the TANITA HR-001 portable altimeter. Waist–hip ratio (WHR) was determined, based on the waist circumference (WC; cm) and hip circumference (HC; cm); both of these parameters were measured by a certified centimeter. BMI (kg/m^2^) was calculated based on the values of body height and BWT of every respondent.

### 2.4. Cardiovascular Parameters

VaSera device (Fukuda Denshi, Japan) was used for the estimation of cardio-ankle vascular index (CAVI) and ankle-brachial Index (ABI), which can be calculated as a ratio of systolic blood pressure (SP) at the ankle to the SP in the arm. CAVI is a parameter based on PWV, but unlike the PWV, it is based on the condition of the whole cardiovascular system. The formula used for CAVI calculation usually takes the following form: CAVI = constant − a × (ln of systolic blood pressure/diastolic blood pressure) × (2 × blood density/pulse pressure) × pulse wave velocity^2^ + constant − b. The electrocardiography (ECG), phonocardiography (PCG) and systolic/diastolic blood pressure measurements (SP/DP) on brachial and ankle arteries were also performed by the VaSera device [13].

PWV (m/s) was measured using the applanation tonometer SphygmoCor (AtCor Medical, Australia) as it is a reliable marker of arterial stiffness. The measurements were performed on the radial artery of respondents’ dominant hand and on their ipsilateral carotid artery. PWV was calculated as a ratio of the distance traveled by the pulse wave in both directions and the difference of transit times in radial and carotid arteries. The distance was measured from the suprasternal notch to the common carotid artery, and from the suprasternal notch to the radial artery. Transit times were calculated as the interval from the R wave on the ECG curve to the steep increase of the pulse wave curve in both arteries. All the measurements were calibrated using blood pressure values obtained from the oscillometric automated measurement on the ipsilateral brachial artery (Omron HEM, 907). In addition, variability of PWV (PWV_V_) was measured in each respondent [14].

The peripheral pulse wave curve was obtained, and SP and DP (mmHg) were estimated using the aforementioned applanation tonometry. Using the SphygmoCor Px system, the measured peripheral pulse wave was further converted into the central aortic pulse wave, and the following parameters were calculated from this central (aortic) curve: augmentation pressure (AP; mmHg), and augmentation index (AI, %). These parameters were standardized to a heart rate of 75/min and to the pulse height [15,16].

The following heart function parameters were estimated from the central pulse wave curve: tension time index (TTI) as the area under the systolic part of the pulse curve, diastolic time index (DTI) as the area under the diastolic part of the pulse curve, and subendocardial viability index (SVI, %) as the ratio of diastolic time index to tension time index. All parameters obtained from the pulse wave analysis are represented in Figure 1.

### 2.5. Statistical Analysis

We used the program Statistica 14.0.0.15 (StatSoft, TIBCO Software Inc., Palo Alto, CA, USA) for the statistical analysis. Given the non-Gaussian distribution, we used the median and lower-upper quartile. First, we used the Wilcoxon signed-rank test to find differences between the measurements I. and II. inside of each of the age groups. The nonparametric Mann–Whitney U test was used for the comparison of all parameters between the sexes.

## 3. Results

### 3.1. Anthropometry

#### 3.1.1. Anthropometric Comparison of the Whole Cohort

The results of nutritional state assessment of the whole group are represented in Table 2. All parameters are above the physiological range. Statistically significant differences between measurements I. and II. were observed in all parameters except the WHR. Higher WC in measurement II. is given by the use of the median, while the Q1 and Q3 values suggest that the whole data set of the II. measurement is shifted to lower values. When comparing males and females, we found significant differences in the following parameters: WC_I_ (90 vs. 82 cm; *p* < 0.05), WHR_I_ (0.89 vs. 0.85; *p* < 0.05), WC_II_ (89 vs. 79.5 cm; *p* < 0.01), WHR_II_ (0.88 vs. 0.82; *p* < 0.01). These significant differences reflect the differences between the human sexes.

#### 3.1.2. Anthropometric Comparison of Sexes within the Age Groups

While comparing males and females, we did not find statistically significant differences in any parameters in the age group 9–11years. In the remaining age groups, we found significant differences in the following parameters:Age group 12–14 years: BWT_I_ (79.9 vs. 67.5 kg; *p* < 0.05), BMI_I_ (30.7 vs. 24.9 kg/m^2^; *p* < 0.05), BMI_II_ (28.9 vs. 23.1 kg/m^2^; *p* < 0.05), WHR_II_ (0.9 vs. 0.8; *p* < 0.01).Age group 15–17 years: Fat_I_ (30.3 vs. 45.7%; *p* < 0.01), Fat_II_ (28 vs. 43.8%; *p* < 0.01), WHR_II_ (0.9 vs. 0.8; *p* < 0.05).

Figure 2 and Figure 3 show BWT and BMI respectively in relation to their percentile graphs for the Czech pediatric population.

Table 3 shows the percentage of fat tissue in both sexes in all age groups, as well as the physiological ranges of fat tissue percentage for children of respective ages.

### 3.2. Cardiovascular Parameters

#### 3.2.1. Comparison of Cardiovascular Parameters of the Whole Cohort

The values of all measured cardiovascular parameters are represented in Table 4. None of our respondents suffer from obesity-related hypertension. Statistically significant differences between the measurements I. and II. were observed in the following parameters: DP, PWV, ABI, TTI, SVI. Comparison of other parameters yielded no significant results. While comparing males and females, we did not find statistically significant differences in any of the parameters except for the SP (SP_I_: 110 vs. 105 mmHg; *p* < 0.01 and SP_II_: 109 vs. 104 mmHg; *p* < 0.05). The difference is most likely caused by short-term variability of blood pressure and baroreflex sensitivity [17].

#### 3.2.2. Comparison of Cardiovascular Parameters of Both Sexes within Every Age Group

The comparison of male and female respondents within every age group did not show any statistically significant differences.

#### 3.2.3. Comparison of Cardiovascular Parameters between the Measurements within Every Age Group

While comparing the changes of parameters between the measurements I. and II. inside every age group, we found significant differences in the following parameters: PWV (parameter is represented in Figure 4), ABI (parameter is represented in Figure 5), and SVI (parameter is represented in Figure 6).

## 4. Discussion

### 4.1. Anthropometry

In our study we used BWT, BMI, percentage of fat tissue, WC, and WHR as parameters for the evaluation of the nutritional state of our respondents. Because anthropometric parameters are sex- and age-dependent, we compared males and females separately in each age group except for the whole cohort comparison. All studied parameters exceeded the 97th percentile in both males and females in each age group. Despite significantly decreased body weight, BMI, fat, and waist circumference after the one-month-long stay in the children’s hospital, our respondents did not reach the physiological range for any of the above mentioned parameters [10].

A comparison between males and females shows the influence of puberty on selected anthropometric parameters. Puberty onset, which leads to the elevation of sex hormones, directly correlates with obesity, while stronger correlation is observed in females [18,19]. The exact mechanism behind the influence of puberty on obesity has not been explained yet; it seems to be of multifactorial etiology, with not only hormonal factors (estrogens, testosterone, leptin), but also environmental factors (family, sociocultural, agroalimentary, fashion) having a significant impact on the resulting nutritional state [20,21]. The comparison of males and females showed different dynamics of obesity development between the sexes, as was proposed by this meta-analysis [22].

Hormonal changes during puberty cause proliferation and modified redistribution of fat tissue: estrogens increase fat deposition in all subcutaneous fat depots, whereas testosterone stimulates the proliferation of visceral fat [23]. Our data confirmed this, as we observed consistently higher WC in male respondents than in females, across all age groups.

Age group 9–11 years does not show any significant differences between males and females in all studied anthropometric parameters—according to Professor J. M. Tanner, children in this age group are in Tanner stages 1 and 2 [24]. The next Tanner stages differ between the sexes, and therefore nonlinear BMI growth can be observed in both sexes, but in different ages; in our study, we observed a drop in BMI in females aged 12–14 years and in males aged 15–17 years, which corresponds to the Tanner stages for body height development.

### 4.2. Cardiovascular Parameters

Arterial hypertension poses an important public health problem, especially in connection to childhood obesity [25]. The rising prevalence of childhood obesity is leading to increasing numbers of obesity-related cardiovascular diseases in children; moreover, it leads to a higher risk of cardiovascular morbidity and mortality in adulthood [26]. It was shown that early signs of cardiovascular dysfunction could be a result of excess fat mass independently, without other comorbidities such as dyslipidemia and insulin resistance [27]. Early detection of vascular dysfunction is essential to identify individuals at risk of subsequent cardiovascular morbidity [28]. Our results for PWV showed significant decreases between the measurements I. and II. in all age groups. Similar results were found in the following study [29]. The significantly decreased DP and carotid-radial PWV in the measurement II., together with no significant changes of CAVI, imply that most of the vascular changes occurred in the peripheral part of the arterial tree. No observed changes in AP and AI also points towards the periphery as the source of arterial stiffness changes. Other studies with a similar aim, which are represented in this meta-analysis carried out by Lee D. Hudson, also assessed the central arterial stiffness—they provide contradictory results, as they show increased, decreased and also unchanged central PWV in obese children, in a direct comparison with healthy controls [30]. Their results can be affected by the influence of physical development during childhood and adolescence; the speed and timing of physical development are altered in obese children, who seem to undergo certain developmental processes earlier than their healthy peers [26,31].

Changes in sympathovagal balance with the decreasing of vagal activity are closely associated with the development of cardiovascular diseases and increased mortality [32]. Generally, sympathovagal activity is analyzed by a heart rate variability examination, but it was also shown that PWV_V_ could be used as a marker of autonomic nerve system activity [33]. Our results do not show significant differences in PWV_V_ between measurements I. and II., which allows us to suggest that in spite of changes in PWV, sympathovagal activity is not impaired.

Overweight and obesity in children and adolescents are associated with impairment of cardiac structure and function [34]. High metabolic activity of excess fat tissue leads to increases in total blood volume and cardiac output, while at the same time stiffer arteries increase the afterload, and thus adaptive processes lead to remodeling and impairment of cardiac function [35]. Several studies have focused on left and right ventricular function in overweight and obese children, where an examination of heart function includes not only echocardiography with strain imaging, but also other imaging techniques [36,37,38]. All of the cited studies show impaired left or right systolic/diastolic heart function in obese children in comparison to the non-obese control group [38,39].

In our project, we chose SVI for the analysis and evaluation of the diastolic heart function. SVI could be used as an indicator of the effectiveness of myocardial oxygenation, which occurs exclusively during the diastole, and changes in this parameter imply possible predispositions of the left ventricle to diastolic dysfunction. Moreover, SVI can be used to predict even subclinical changes of the myocardial tissue [40]. Our respondents showed significantly higher SVI after their one-month long stay in the children’s hospital. In the age groups 9–11 years and 12–14 years, the initial SVI was on the lower border of its physiological range. Similar SVI values were found in a Slovenian pilot project, where SVI was analyzed not only in obese children, but also in children with hypertension and hypercholesterolemia [41]. The long-term progression of cardiac function impairment has not been thoroughly studied yet, nor whether it has health-related outcomes in adulthood. H. Yang shows unchanged heart function in obese adult who were obese in childhood, and significant deterioration in obese adults who were not obese in childhood [42]. Lalan’s study showed significant correlation between higher BMI in childhood and lower SVI in adulthood [43].

Obesity is often accompanied by atherosclerosis and peripheral artery disease (PAD); their connection has, however, been studied and described mostly in respondents over 40 years of age [44]. Asymptomatic development of PAD combined with belated diagnosis can lead to exacerbation of PAD and irreversible changes to the arteries. The most-used parameter for non-invasive assessment of PAD is currently the ABI [45]. Throughout the whole cohort and within every age group, we observed significantly increased ABI after the one-month stay in the children’s hospital. The results of our study also show borderline ABI values for the 9–11 years-of-age group during the first measurement. ABI values below 0.90 are an indicator of possible PAD; this disease is, however, rarely observed in young people, and therefore the results of our respondents are not worrying, at least in the short term [46,47].

### 4.3. Strengths and Limitations

While published data from other relevant studies always describe only the peripheral or only the central part of the cardiovascular system, we merged both of these approaches in order to gain additional information and better understanding of the changes happening in the cardiovascular system. This allowed us to describe the state of the cardiovascular system of our respondents in a more complex manner.

Despite the generally solid design of this study, it was impossible to create large enough age-based subgroups, since the number of participants was limited. This could unfortunately lead to reduced significance of the results of certain parameters This seems to be the biggest downside of our project, although we believe it can still be compared to similar studies.

## 5. Conclusions

In this study, we assessed a number of anthropometric and cardiovascular parameters. One-month-long weight-loss therapy led to significant improvements in measured parameters in the whole cohort, as well as in every age subgroup, with stronger statistical significance in the anthropometric parameters, in which we observed a significant improvement immediately after the therapy. The results of this study show that obesity has a mostly negative impact on the whole cardiovascular system, mainly damaging its peripheral part (DP, PWV). We also found a negative impact of obesity on the diastolic heart function of the respondents, which was represented by SVI. These changes were not likely to be found by other clinically used methods of diastolic heart function assessment, as SVI allows us to detect preclinical changes of heart function. SVI could therefore be used in the future as a predictor of cardiovascular disease manifestation. All of the above mentioned changes could likely lead to negative health consequences in the patients’ adulthood. Based on the limited amount of published data, further research in this field is necessary. We are considering a follow-up study with this specific cohort of pediatric respondents and with healthy respondents as a control group.

## Figures and Tables

**Figure 1 children-09-01610-f001:**
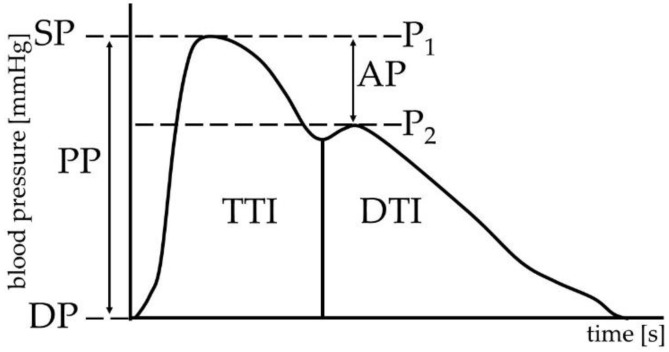
Central pulse curve. SP—systolic blood pressure, DP—diastolic blood pressure, PP—pulse pressure, P_1_—pressure at the peak systolic (forward) pulse wave, P_2_—pressure at the peak of reflected (backward) pulse wave, AP—augmentation pressure, TTI—tension time index, DTI—diastolic time index.

**Figure 2 children-09-01610-f002:**
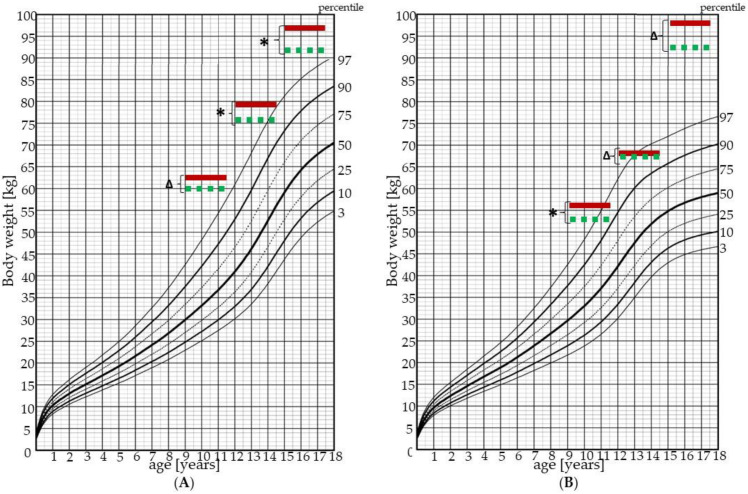
Body weight (BWT) in measurements I. and II. for males (**A**) and females (**B**). Solid red line represents measurement I., which was made at the beginning of the one-month stay in the children’s hospital, dashed green line represents measurement II., made at the end of the one-month stay in the children’s hospital. *—*p* value is less than 0.01; Δ—*p* value is less than 0.05; *p* value—statistical evaluation of the differences between the measurements I. and II., Wilcoxon signed-rank test (paired test).

**Figure 3 children-09-01610-f003:**
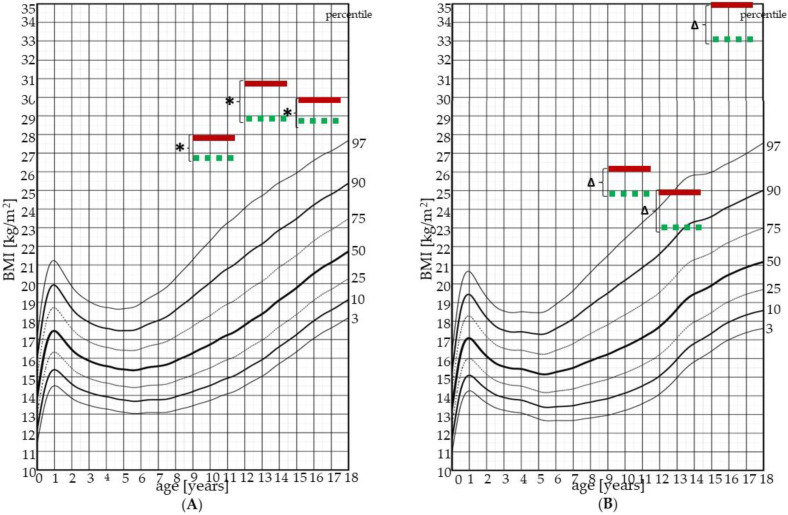
Body mass index (BMI) in measurements I. and II. for males (**A**) and females (**B**). Solid red line represents measurement I., made at the beginning of the one-month stay in the children’s hospital, dashed green line represents measurement II., made at the end of the one-month stay in the children’s hospital. *—*p* value is less than 0.01; Δ—*p* value is less than 0.05; *p* value—statistical evaluation of the differences between the measurements I. and II., Wilcoxon signed-rank test (paired test).

**Figure 4 children-09-01610-f004:**
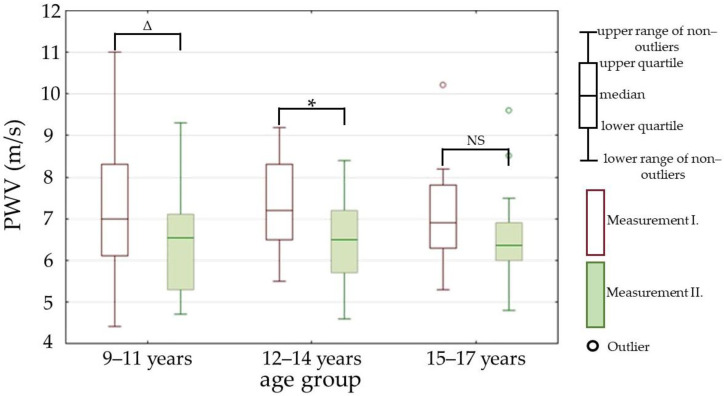
Pulse wave velocity (PWV) in measurements I. and II. Measurement I.—at the beginning of the one-month stay in the children’s hospital; measurement II.—at the end of the one-month stay in the children’s hospital; *—*p* value is less than 0.01; Δ—*p* value is less than 0.05; NS—non-significant; *p* value—statistical evaluation of the differences between the measurements I. and II., Wilcoxon signed-rank test (paired test).

**Figure 5 children-09-01610-f005:**
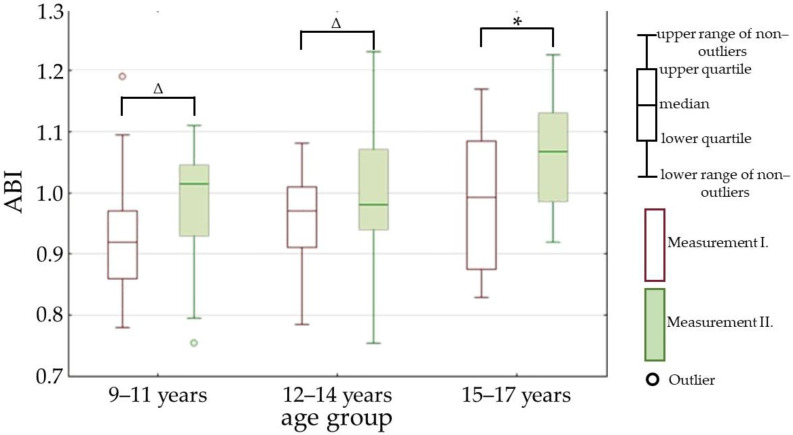
Ankle-brachial index (ABI) in measurements I. and II. Measurement I.—at the beginning of the one-month stay in the children’s hospital; measurement II.—at the end of the one-month stay in the children’s hospital; *—*p* value is less than 0.01; Δ—*p* value is less than 0.05; *p* value—statistical evaluation of the differences between the measurements I. and II., Wilcoxon signed-rank test (paired test).

**Figure 6 children-09-01610-f006:**
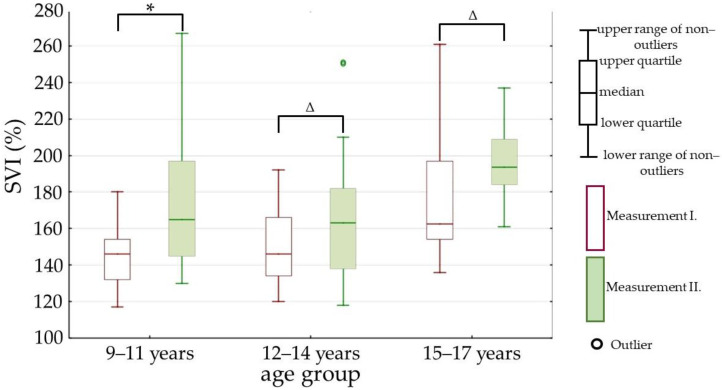
Subendocardial viability index (SVI) in measurements I. and II. Measurement I.—at the beginning of the one-month stay in the children’s hospital; measurement II.—at the end of the one-month stay in the children’s hospital; *—*p* value is less than 0.01; Δ—*p* value is less than 0.05; *p* value—statistical evaluation of the differences between the measurements I. and II., Wilcoxon signed-rank test (paired test).

**Table 1 children-09-01610-t001:** The basic characteristics of the respondents.

		BWT (kg)	BMI (kg/m^2^)	SP (mmHg)	DP (mmHg)
Whole cohort		76 (63–89)	28.6 (25–31.6)	126 (117–139)	70 (64–75)
9–11 years	Males	63 (60–67)	27.9 (27.2–29)	116 (115–123)	65 (63–74)
	Females	55.7 (50–74.2)	26.1 (24.4–30.3)	115 (112–127)	66 (64–75)
12–14 years	Males	80 (72–91)	30.7 (26–31)	137 (130–145)	70 (65–74)
	Females	68 (66–78)	24.9 (24–30)	124 (122–125)	66 (66–74)
15–17 years	Males	96.6 (77.8–105.5)	30 (27.3–33)	142 (133–146)	72 (70–75)
	Females	98.4 (92–100.5)	35.4 (33.4–38.9)	136 (123–138)	80 (73–80)

Parameters are expressed as median (lower quartile–upper quartile). BWT—body weight; BMI—body mass index; SP/DP—systolic/diastolic blood pressure.

**Table 2 children-09-01610-t002:** Evaluation of nutritional state—common data set.

Parameter	Measurement I.	Measurement II.	*p* Value
BWT (kg)	75.5 (63.4–89.4)	72.6 (61.8–86.9)	*p* < 0.01
BMI (kg/m^2^)	28.6 (25.4–31.6)	26.9 (24.2–30.3)	*p* < 0.01
Fat (%)	36.9 (32.2–42.3)	34.5 (28.9–39.2)	*p* < 0.01
WC (cm)	87.5 (82.5–95.5)	88 (80–92.5)	*p* < 0.01
HC (cm)	101.5 (95.5–111.5)	99.5 (95–109.5)	*p* < 0.01
WHR	0.9 (0.8–0.9)	0.9 (0.8–0.9)	NS

Parameters are expressed as median (lower quartile–upper quartile). Measurement I.—at the beginning of the one-month stay in the children’s hospital; Measurement II.—at the end of the one–month stay in the children’s hospital; BWT—body weight; BMI—body mass index; WC—waist circumference; HC—hip circumference; WHR—waist to hip ratio; *p* value—statistical evaluation of the differences between measurements I. and II., Wilcoxon signed-rank test (paired test), NS—non–significant.

**Table 3 children-09-01610-t003:** Percentage of fat tissue.

		Male	Female	*p* Value (M vs. F)
9–11 years	Physiological range	12–23%	16–28%	
	Measurement I.	40.5 (36.4–43.5)	37 (36.2–38.9)	NS
	Measurement II.	38.5 (36.6–40.8)	35.4 (32–37.6)	NS
	*p* value (I. vs. II.)	NS	*p* < 0.05	
12–14 years	Physiological range	11–22%	16–29%	
	Measurement I.	35 (31.6–41.3)	35.4 (31.5–38.7)	NS
	Measurement II.	30.1 (27.4–35.8)	30.9 (26.4–37.5)	NS
	*p* value (I. vs. II.)	*p* < 0.01	NS	
15–17 years	Physiological range	10–20%	16–30%	
	Measurement I.	30.5 (23.7–36.1)	45.7 (43.3–47.1)	*p* < 0.05
	Measurement II.	28 (24.5–34.2)	43.8 (39.6–45.5)	*p* < 0.05
	*p* value (I. vs. II.)	NS	NS	

Parameters are expressed as median (lower quartile–upper quartile). Measurement I.—at the beginning of the one-month stay in the children’s hospital; measurement II.—at the end of the one-month stay in the children’s hospital; *p* value—statistical evaluation of the differences between measurements I. and II.—Wilcoxon signed-rank test (paired test), or statistical evaluation of the differences between males and females—Mann–Whitney test, NS—non-significant.

**Table 4 children-09-01610-t004:** Cardiovascular parameters.

Parameter	Measurement I.	Measurement II.	*p* Value
SP (mmHg)	108 (100–117)	107 (100–112)	NS
DP (mmHg)	64 (60–71.5)	62 (58.5–69)	*p* < 0.05
AP (mmHg)	0 (−6–7)	0 (−8–4)	NS
AI (%)	97 (91.5–104)	98 (94–106)	NS
PWV (m/s)	7.1 (6.3–8.2)	6.5 (5.7–7.2)	*p* < 0.01
PWV_V_ (m/s)	0.6 (0.5–0.7)	0.6 (0.4–0.8)	NS
CAVI	4.3 (3.6–5)	4.3 (3.8–4.9)	NS
ABI	0.96 (0.88–1.01)	1.02 (0.94–1.07)	*p* < 0.01
TTI	1910 (1770–2075)	1710.5 (1499.5–1888)	*p* < 0.01
DTI	2758.5 (2618.5–3064)	2934.5 (2756–3127)	NS
SVI (%)	151 (136–166.5)	173.5 (146.5–198)	*p* < 0.01

Parameters are expressed as median (lower quartile–upper quartile). Measurement I.—at the beginning of the one-month stay in the children’s hospital; Measurement II.—at the end of the one–month stay in the children’s hospital; SP/DP—systolic/diastolic blood pressure; AP—augmentation pressure; AI—augmentation index; PWV—pulse wave velocity (carotid–radial PWV); PWV_V_—variability of PWV; CAVI—cardio–ankle vascular index; ABI—ankle–brachial index; TTI—tension time index; DTI—diastolic time index; SVI—subendocardial viability index; *p* value—statistical evaluation of the differences between the measurements I. and II., Wilcoxon signed-rank test (paired test), NS—non-significant.

## Data Availability

The data presented in this study are available on request from the corresponding author.

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
