# Peer review of "Changes in Nutritional State and Cardiovascular Parameters in Alimentary Obese Children after a Month-Long Stay in Children’s Treatment Center"

_children, 2022, doi:10.3390/children9111610_

Round 1
Reviewer 1 Report
Introduction: Overall, there should be information about what is alimentary obesity in the study setting since, as of now, it is focused more on the consequences.
- Lines 33, 36-37 should include a reference and may be improved with specific examples
- First part of Line 42 may be moved to another part for further context since adolescence and childhood are not connected in this paragraph
-Alimentary obesity can be defined in methods and the title / other sections can be refer only to obesity
Methods: It needs to be improved, specifying the study design at the beginning. Should you consider organize the section as follows: study setting/participants (clinic for obesity treatment, and relevant information to understand the treatment), study design - longitudinal?, specifying the participants selection criteria, follow-up times, data collection for exposure/outcomes and assessments at follow-ups, loss-to-follow-up? confounders measured, ethics.
Results: Readers would benefit from the participants characteristics. Age, sex, anthropometry, and other covariates of interest distribution at baseline (Measurements I).
- Lines 150-159 are difficult to follow, I would suggest to edit it to a narrative way rather than bullet points, and refer to tables (such as table2) with relevant results
The results section organization could be improved so the discussion is easier to understand.
Discussion: Needs to be improved and not only contain the results expanded
- Lines 227-232 should contain a summary of the main findings and the WHO information could be relevant for the introduction section
- Lines 234 is a standard definition that should be explained earlier to understand the whole study, instead, the authors could start directly with the problem of limitations of BMI for fat measurements in the body
-The discussion does not provide Strengths or limitations section. For example, this is a small sample size and even smaller with the age classification, how would that influence certain conclusions. However, there are not too many studies like this, why is it important? I think the answer to these questions should be included, as the authors also mentioned that they are considering further studies within this cohort.
- It was a little difficult to follow the key findings and comparison between groups of interest. The authors could consider an age group flow and discuss the results in that order since it appears to be a biological change depending on the age
Author Response
Dear reviewer,
Thank you for your kind comments and suggestions. We tried to improve the article according to your review report and you can see the final result of our effort in the attachment. We rewrote the introduction using parts of text from the original discussion part and included more information about alimentary obesity. In the methods part, we more thoroughly specified the design of the study. We also added characteristics of our respondents, including their anthropometric and cardiovascular parameters. After an intense debate, the authors decided to leave the percentile graphs as a representation of the results, instead of switching to a set of tables; since we already created the tables, I am sending you an example of a table containing all the anthropometric parameters for one of the age groups. If you insist on changing the results representation, please let me know.
|
|
|
|
Male |
Female |
P value (M vs. F) |
|
9 – 11 y |
BWT (kg) |
Measurement I. |
62.7 (59.9–66.9) |
55.7 (49.5–74.2) |
NS |
|
Measurement II. |
60.4 (56.3–63.7) |
53.5 (47.3–70.5) |
NS |
||
|
P value (I. vs. II) |
p<0.01 |
p<0.05 |
|
||
|
BMI (kg/m2) |
Measurement I. |
27.7 (27.2–29) |
26.1 (24.4–30.3) |
NS |
|
|
Measurement II. |
26.8 (24.7–27.6) |
24.8 (23.3–27.1) |
NS |
||
|
P value (I. vs. II) |
p<0.01 |
p<0.05 |
|
||
|
Fat (%) |
Measurement I. |
40.5 (36.4–43.5) |
37 (36.2–38.9) |
NS |
|
|
Measurement II. |
38.5 (36.6–40.8) |
35.4 (32–37.6) |
NS |
||
|
P value (I. vs. II) |
NS |
p<0.05 |
|
||
|
WC (cm) |
Measurement I. |
86 (85–92) |
80.5 (76–89) |
NS |
|
|
Measurement II. |
87 (83–90) |
77.5 (76.5–88) |
NS |
||
|
P value (I. vs. II) |
NS |
NS |
|
||
|
HC (cm) |
Measurement I. |
97 (91–101) |
91.5 (89–100.5) |
NS |
|
|
Measurement II. |
96 (91–100) |
91 (88–96.5) |
NS |
||
|
P value (I. vs. II) |
p<0.05 |
NS |
|
||
|
WHR |
Measurement I. |
0.89 (0.87–0.93) |
0.87 (0.8–0.9) |
NS |
|
|
Measurement II. |
0.87 (0.85–0.95) |
0.87 (0.8–0.9) |
NS |
||
|
P value (I. vs. II) |
NS |
NS |
|
Parameters are expressed as median (lower quartile–upper quartile). Measurement I.—at the beginning of the one–month stay in the children’s hospital; Measurement II.—at the end of the one–month stay in the children’s hospital; BWT—body weight; BMI—body mass index; WC—waist circumference; HC—hip circumference; WHR—waist to hip ratio; P value — statistical evaluation of the differences between measurements I. and II., Wilcoxon signed–rank test (paired test), or statistical evaluation of the differences between males and females—Mann–Whitney test, NS—non–significant.
The discussion part was also corrected and some paragraphs were used in the introduction instead. Additionally, we added the “Strengths and limitations” part. We divided the whole discussion into several parts according to discussed parameters (anthropometry, cardiovascular system). You were right about the age-related changes in cardiovascular parameters, which is the reason why we divided the whole group into subgroups; our discussion is, however, focused on each parameter separately, because the reason of the changes is the same.
I would like to thank you once again for your comments and suggestions.
Best regards
Ksenia Budinskaya
Physiology department
Masaryk university, Brno, CZ

Reviewer 2 Report
Dear Authors,
Thank you for presenting your interesting study. The manuscript is nicely written, but it can be improved.
First, I have some general comments:
- English should be improved, I suggest that a native English speaker reviews the manuscript
- if there is more then one reference, they should be put togheter in the brackets, for example [4, 5]
- generally, references are not listed according Children instructions, please correct - for example in the first reference you have one author and then et al already, with use of abbreviations such as roč, č, říj?
Next, some suggestions for each segment are listed below.
Introduction:
- I suggest to define prevalence for pediatric population in general, not only 6-9 years old, since your responders are also older children
- lines 48-53 could be better described, maybe with the use of crucial word - atherosclerosis
- define the aim of the study more clearly
- since the special issue is about hypertension, state more clearly how arterial stiffness is associated with elevated blood pressure (I see the association, but all readers may not)
Methods:
- describe the function and cardiovascular parametrs derived from VaSera device more in detail
Results: nicely presented
- did any of participants had obesity-related hypertension?
Discussion: nicely written
- line 235 however is written twice
- looking back to results, comment on why is WC higher (Table 1) after losing weight problem?
- did participants have problems with one month long stay? How did they perceive it? It is very unusual to have a month-long stay
- if there is enough patients with hypertension, it would be good to compare them separately to the whole group, since the emphasis of the issue is on hypertension
Conclusions:
- state more clearly which cardiovascular parameters improved during the study and how this impacts further cardiovascular risk (e.g. hypertension)
- I suggest to rewrite the conclusions with focus more on the results; we already know that obesity negatively impats cardiovascular helath, I suggest emphasizing that after one month long weight-loss program many of the cardiovascular (name which) parameters improved significantly
Kind regards!
Author Response
Dear reviewer,
Thank you for your kind comments and suggestions. We tried to improve the article according to your review report and you can see the final result of our effort in the attachment. We rewrote the introduction, which is now about the whole age range, not only about the age group of 6–9 y. We also included and specified various VaSera parameters. Also, we rewrote the conclusions section, specified the impaired cardiovascular parameter and mentioned the changes of the parameters after the one month long stay in the hospital.
I would also like to explain the one month long stay in children hospital. It is the most common duration of stay for children with different health issues (bronchial asthma, skin diseases, obesity and so on). This stay is paid for by insurance companies and everyone can use this opportunity before they reach 18 years of age. Children basically do not have problems with staying in the hospital, but in case of any problems, there is always a psychologist available for the young patients. It should also be mentioned that this hospital is rather special because it is well suited for the young patients and is situated in a small chateau in the nature. The children usually know each other because they visit this hospital once every year. The main problem with the compliance of the patients is, however, mostly the incompetence of their parents, because when the children are at home, the parents do not tend to adhere to the individual meal plans, they do not support the children in exercising and so on. So in conclusion, one month is usually enough for forming healthy habits, but not enough to keep them strong.
Now about the obesity-related hypertension – we do not have respondents with obesity-related hypertension in our cohort. To be exact, pediatric patients with obesity-related hypertension have a different treatment plan and therefore were not suitable for participation in our research. We are however planning to include such respondents in our project.
As a last point, I would like to mention the WC results. In our article we used median and lower/upper quartiles. In spite of the median being higher after the weight loss therapy, we could see that the whole data set was shifted towards lower values. In case of average values for this parameter, the comparison of measurements I. and II. looks like this: 89.5±11.1 cm vs 87.5±9.6 cm.
Thank you very much for comments and suggestions
Best regards
Ksenia Budinskaya
Physiology department
Masaryk university, Brno, CZ

Reviewer 3 Report
Dear authors
The paper is of interest because it deals with a very important health problem, especially since the study population is adolescents. Also, making comparisons by sex is very pertinent.
I have several questions or comments
Major issues
Although the results are very interesting, I am concerned about the sample size. If a sample size study was done, I think it would be appropriate to state this in the paper.
The psychological and behavior change therapies are hardly defined and unclear.
I recommend that this project be continued by looking for a control group to compare if indeed, the intervention is efficient.
I would also like to know, or it should be stated in the text, why these age groups have been taken. If there is no criterion for taking these groups, I think it is better not to make them and thus have more statistical power.
The comparison by sex should be made in the first measurement, in the second. In addition, it would be good to study how men have changed by comparing male/male and female/female in measurement I and II.
I also propose, if possible, to quantify this change by calculating the effect size (Cohen's d or Hedges' g).
Minor issues
The first paragraph of the Discussion section is an Introduction paragraph, therefore it should either be removed from the text or moved.
Author Response
Dear reviewer,
thank you for your kind comments and suggestions. We tried to improve the article according to your review report and you can see the final result of our effort in the attachment. We rewrote the introduction and it now includes related parts of the text from the original discussion. The methods part now contains information about cognitive–behavioral therapy. Unfortunately, we do not have the opportunity to provide Cohen`s d / Hedges` g test, but we appreciate this comment, and we will use that test in our future projects. We used the data of 60 respondents in this pilot project since these were all the respondents of whom we had complete data sets for the parameters described in this article. We decided to choose respective parameters because they describe changes in central and peripheral parts of the vascular system as similar projects focused on the central or peripheral part, but (to our knowledge) not on both at the same time. We also created 3 age-based subgroups because of the age-dependent changes in the cardiovascular system. Currently, we continue with the project and it now also includes children with obesity-related hypertension and a control group which consists of healthy children.
I would once again like to thank you for your comments and suggestions.
Best regards
Ksenia Budinskaya
Physiology department
Masaryk university, Brno, CZ

Round 2
Reviewer 1 Report
-Thank you for considering the suggestions provided. The current version has all the elements to guide the reader through the research conducted. There are some minor suggestions below:
-The english writing style in the new section of strengths and limitations could be improved and limitations about the lack of a priori power calculation should be mentioned.
-Conclusion should emphasize the summarized positive changes found after the weight loss treatment. It should be aligned with the main aim of the study. I suggest to reorganize it in a way that you show first the conclusion related to the main aim and then, other interesting/major findings.
Author Response
Dear reviewer,
thank you for your feedback after the first round of revisions. In the current version of our manuscript, we improved the writing style of the newly added “Strengths and limitations” section. We also reorganized the conclusions section in order to highlight the positive aspects of weight-loss therapy even better – it now consists of a part summarizing the general results of the therapy and a part specifically aimed at the cardiovascular consequences of the therapy. You can find the current version of the manuscript attached to this message.
Best regards
Ksenia Budinskaya

Reviewer 2 Report
The authors adjusted the manuscript accordingly and answered my questions.
Kind regards,
Author Response
Dear reviewer,
We would like to thank you for your comments and suggestions in the first round of revisions. We really appreciate them as we believe they significantly helped us to improve our manuscript.
Best regards
Ksenia Budinskaya
Reviewer 3 Report
Thank you very much for answering all my questions and I recommend continuing to work on this project because it is very interesting.
Author Response

(The authors gave the same response as above.)
